# The Influence of Different Auditory Stimuli on Attentiveness and Responsiveness in Road Traffic in Simulated Traffic Situations

**DOI:** 10.3390/ijerph17249226

**Published:** 2020-12-10

**Authors:** Wolfgang Welz, Susanne Voelter-Mahlknecht, Christian Große-Siestrup, Geraldine Preuß

**Affiliations:** 1Evangelisches Krankenhaus Ludwigsfelde-Teltow, Akademisches Lehrkrankenhaus der Charité, Albert-Schweitzer-Straße 40-44, 14974 Ludwigsfelde, Germany; 2Institut für Arbeitsmedizin der—Universitätsmedizin Berlin, Seestraße 73, 13347 Berlin, Germany; susanne.voelter-mahlknecht@charite.de (S.V.-M.); christian.grosse-siestrup@charite.de (C.G.-S.); 3Gesellschaft für Leben und Gesundheit mbH, Rudolf-Breitscheid-Straße 36, 16225 Eberswalde, Germany; geraldine.preuss@gmx.de

**Keywords:** auditory stimuli, deflection, responsiveness, road safety

## Abstract

The use of portable media has become an integral part of our increasingly mobile society. The use of digital audio books is also growing steadily in Germany. The connection between the psychological effect of music of different volumes and rhythms and the change in reaction in road traffic with a corresponding increase in risk behavior, especially when driving, has already been proven in previous studies. Only a few studies are available on the effects of listening to radio plays on reaction behavior and concentration in road traffic as well as on risk behavior among pedestrians and cyclists. In the present study, we have investigated the influences of pop music and a radio play on reaction behavior and thus driving ability during the execution of a traffic psychological test series from the “Wiener Test System”. The central topic deals with the performance of the test subjects in the individual tests. Conclusions are drawn on the reaction behavior and concentration during participation in road traffic and thus the risk of distraction and possible increased risk of accidents. Studies on the influence of auditory stimuli and their effects on concentration and reaction during participation in traffic are of great interest from the point of view of traffic psychology and occupational medicine, since a reduction in the risk of accidents can increase general traffic safety and lead to a decrease in sick leave and therefore fewer absences from work.

## 1. Introduction

Today, especially in cities, we are exposed to a variety of noises which, taken together, can have demonstrably negative psychological and physical effects on the body. In particular, young people put additional strain on their hearing by using portable MP3 music devices and mobile phones. This can pose a risk while simultaneously participating in road traffic, regardless of whether one turns up the car stereo to full volume or is on the road as a cyclist or pedestrian with loud music and headphones turned up. After sight, hearing is the most important requirement for safe participation in traffic. Through good auditory perception, relevant information about the speed and direction of movement of other road users can be adequately obtained and processed.

In 2019, traffic deaths in Germany decreased by 7 percent to 3046 persons killed compared to 2018. This continues a trend of decreasing deaths between 2016 and 2019, after an increase in the number of road users killed between 2013 and 2015. The proportion of young adults aged between 18 and 24 years who were killed remains high, accounting for 12 percent of all road deaths. In relation to one million inhabitants in their age group, young adults are the most vulnerable age group, with 58 fatalities [1]. One reason for the high percentage of deaths in these age and person groups in particular may be the increasing trend towards distraction through the use of smartphones and MP3 players, which can contribute significantly to visual and auditory distraction from road traffic.

In order to understand the effects of auditory influences on response behavior, we need to understand the importance of sensory perception of hearing as a complex brain performance process of the reception, processing and reaction of acoustic stimuli. The physiological significance of hearing results from the interaction of the flow of information and its processing, the influence on our emotions and the fundamental importance for social life. Auditory perception, processing and evaluation is a highly complex brain performance process. The corresponding signal perception and processing takes place in ascending form of the central auditory pathway as so-called “bottom-up processes”, which are increasingly influenced by “top-down processes”—for example, vigilance, attention and memory performance [2].

At the time of our investigations, a recent study was available, which observed a tripling of fatal accidents of pedestrians in the USA in the years 2004–2011, who were proven to have worn headphones and had not noticed approaching vehicles in time [3].

As early as 1984, studies in driving simulators demonstrated the influence of music on driving behavior. Music had a negative influence on reaction time in complex traffic situations. There were also age differences, with higher risk behavior in young drivers [4]. People who listen to relaxing music while driving a car are no worse in their reactions than drivers who do not listen to music [5]. On monotonous routes, it can be useful to simply listen to the radio or use the telephone (hands-free) while driving to avoid fatigue [6]. There may also be an increase in walking speed with fast and loud music, and the effects of volume and speed on heart activity and physical performance [7]. Faster music can increase the speed of walking and increase the willingness to take risks [8]. Other authors also see an influence of different music styles on the cardiovascular system. For example, music with lyrics activates additional brain areas for speech processing, as opposed to purely classical music [9]. The volume of music listened to can also have a significant impact on responsiveness [10]. The presentation of recorded road traffic noise can have the same significant effect [11].

Furthermore, it has been shown that Mozart’s Piano Sonata in D Major (KV 448) can lead to performance-enhancing brain activity [12]. However, it was later found that this effect is independent of what the individual likes better, a piece of music or a passage of text [13]. Language and music are processed by our brain in closely related areas. The more complex sounds we hear, the more we are mentally challenged.

The use of smartphones combines visual and audio distraction like no other medium, with the greatest risk of distraction when writing text messages and listening to music [14]. 

In 2018, a meta-analysis of 41 studies on walking, cycling and driving were evaluated [15]. In most of the studies, the evaluation of the results was considered to be very complex, as distractions can occur at different levels and there can be some accumulation of distractions. As expected, in the majority of cases, visually demanding activities led to altered reaction speeds in almost all groups. Interesting were the negative influences on cognitive behavior, with correspondingly longer reaction times due to demanding telephone conversations. In the majority of cases, the general distraction caused by smartphones has recently been investigated due to the high relevance of the high number of users. So far, there have been few studies specifically on the influence of radio plays on reaction behavior, although radio plays are being consumed by more and more people even while driving a car [16]. During the presentation of radio plays, the braking time can be longer in more complex situations [17]. On the basis of the studies available to date on the subject of auditory influence during road participation, it is possible to determine the influence of different types of rhythm, loudness, text content of what is heard, as well as the age and experience of the consumers and the traffic-related and emotional stressors on psychological and physical reactions. 

### Research Question and Hypotheses

In our study, we consider the effects of different types of sound reinforcement on the concentration ability of participants in road traffic. The aim is to show a correlation between the influence of auditory stimuli and a resulting change in concentration and attention in a road traffic simulation. The central question of our study is whether and to what extent sound reinforcement with a radio play can influence concentration during participation in road traffic. First of all, we are interested in the general connection between the auditory stimulation and the performance in the traffic psychological test. For this purpose, the respective groups will be compared with a control group. Differences with significant deviations of the respective groups in the results allow conclusions to be drawn about a general influence. Furthermore, we are interested in correlations between the type of auditory stimulation and the test result. For this purpose, the groups with music and radio play performance will be compared with each other. Conclusions can be drawn from the different types of acoustic irradiation and the performance in the test. Finally, the question of a possible reduction in the selective perception for the contents of a presented radio play during the simultaneous performance of a traffic psychological test will be addressed. We aim to make statements about the connection between the auditory influence and a resulting change in the performance of the traffic psychological test series.

**Hypothesis** **1.**
*The presentation of a radio play during the performance of a traffic psychological test leads to significantly worse reaction behavior compared to the presentation of music or lack of auditory influence.*


**Hypothesis** **2.**
*The auditory influence during the performance of a traffic psychological test leads to significantly worse performance compared to a control group, regardless of the type of study.*


## 2. Material and Methods

### 2.1. Group Formation

We examined a total of 90 test persons (59 female, 31 male) between 17 and 49 years of age. The average age was 25.2 years. The sex and age distribution of the test persons was random, but the young age average allowed us to make statements for younger road users. The subjects were randomly divided into 3 groups of equal size (*n* = 30). All three groups underwent a traffic psychology test consisting of 5 parts. The control group performed the test without any auditory influence. The music group listened to pop music from the Beatles during the test. The radio play group was presented with a crime thriller radio play. Subsequently, the performance of the groups with music or radio play sound was compared to a control group. The test subjects were volunteers, most of whom were recruited from the psychology course at the Charité Berlin and students from the medical training academy of the Ruppiner Kliniken GmbH (academic teaching hospital of the Charité). Some of the test persons were recruited through advertising for the study. The studies were conducted in the period from 19 April 2011 to 28 October 2011. The publication of the results at this stage is important given the continuing relevance of the subject.

### 2.2. Study Design

We conducted a controlled clinical trial (Controlled Clinical Trial = CCT). The investigations were carried out at the Institute of Occupational Medicine of the Charité Berlin and at the Medical Education Academy of the Ruppiner Kliniken GmbH (academic teaching hospital of the Charité). Using the “Wiener Test System” (WTS), 90 subjects were tested for reactive resilience, orientation, concentration, attentiveness and responsiveness. The “Wiener Test System” (WTS) is considered worldwide as the gold standard for computer-aided psychometric diagnostics and is easy to use even without previous computer knowledge [18]. During the test, the music group was given a selection of Beatles songs from the “Red Album 1962–1966”, released under the label EMI Records Ltd.© 2009. The following songs were performed: Love Me Do, Help, Paperback Writer, A Hard Day’s Night, Can’t Buy Me Love, All My Loving, Ticket to Ride, Drive My Car. During the test, the radio play group was exposed to the chapters 1–5 from the radio play series “Cry of Fear—Metro Feeder 4”, released under the label Marctropolis© 2010. After completing the test, the music group and the radio play group received a questionnaire with questions about the music or speech content of the radio play heard during the test.

### 2.3. Questionnaires

After completing the test, all test persons were given a standardized questionnaire to fill out, developed according to their respective group. The initial part contained 13 general questions, which were identical for all 3 groups. We asked about general listening habits, preferred music styles, any known hearing impairments, etc. In the control group, no further questions were asked. In the music group, 5 additional questions were asked about the music presented during the test. In the radio play group, 9 additional questions were asked on the content of the radio play presented during the test.

### 2.4. Method and Examination Procedure

To carry out the test series, we needed a quiet, closed room as well as a workplace with a computer and a sufficiently large monitor, optimally 19 inches. Figure 1, Figure 2 and Figure 3 show the workplace of the test persons. The software of the “Wiener Test System” (WTS) of Schuhfried GmbH was also used with the corresponding accessories. In our case, these are the standard response panel keyboard and two connected foot pedals. During the test, the subjects were given connected stereo headphones. These were required due to additional acoustic signals in 3 of 5 tests performed, which is why the control group also had to put on the headphones without any further auditory interference. The volume was individually adjusted in the groups, which received music or the radio play in parallel, so that the participants could perceive the signal sounds in parallel despite the music or the radio play.

The “Wiener Test System” (WTS) Traffic is the international standard for driving aptitude diagnostics. It is used to test drivers with conspicuous driving behavior, drivers with possible impaired mental, physical or cognitive abilities and professional drivers who have a particularly high level of responsibility in road traffic. In our study, we use the test set for performance testing according to FEV (driving license regulation) Annex 5 No.2 in a modified form. The test set assesses driving-related ability dimensions and meets the statutory requirements of Germany’s driving license regulations FeV Annex 5 Number 2.

The test set allows economical and exact assessment of the statutorily defined ability dimensions. The traffic psychological investigation can be adapted flexibly to the specific conditions. FEV test set is used to investigate the traffic psychology relevant dimensions of resilience, orientation, concentration and attention performance as well as reaction ability. The test set is characterized by a high degree of economy due to its duration of only around 45 min, while at the same time, the test procedures are highly accurate. For the evaluation of the performance test according to FeV Annex 5 No. 2, the main variable is always used. Table 1 shows the different categories of tests.

For our study, test form S1 (adaptively short), as used in the test battery of the traffic psychology test FeV Annex 5 Number 2, was replaced by test form S4 in the determination test. It offers a combination of free and fixed presentation time. This allows an adaptation to the reaction times of the test persons and also to the subjective feeling of stress. Due to the amount of data collected in the 5 tests, we limit ourselves to the interpretation of the main variables.

### 2.5. Test Forms

The following Table 2 gives a brief overview of the implementation and evaluation of the individual tests.

### 2.6. Biometrics and Statistics

The SPSS Statistics 21 for Windows program (SPSS Inc. an IBM Company^®^, Chicago, IL, USA) was used for statistical analysis. The descriptive specification of the data was followed by testing for normal distribution using the Kolmogorov–Smirnov test and the Shapiro–Wilk test, with significant deviations from a normal distribution in all tests. Therefore, further analysis was performed using non-parametric methods. For this purpose, the Kruskal–Wallis H-Test was used as a rank variance analytical method for comparing several independent samples. Finally, the respective significance was determined. For the evaluation of the questions on the radio play and the music following the test, the Mann–Whitney U-Test was first performed for two independent samples. The null hypothesis was that the difference between the groups was zero, i.e., that the distribution of the score value across the two categories of groups was identical. Thus, it was possible to test whether there was any difference in perception between the music and radio play groups. In order to find out whether the influence of the music or the radio play on the respondents’ reactions is perceived at all, the Wilcoxon Signed Rank Test was then carried out for a sample (total sample). The null hypothesis was that the score value is equal to zero, i.e., that the acoustic disturbance does not have any influence on the perceived reaction behavior.

## 3. Results

### 3.1. Reaction Test

The main variables that we evaluate are the mean reaction time and the mean motoric time. In the descriptive statistics, the radio play group showed a slightly increased mean reaction time of 437 ms compared to the music group with 422 ms and the control group with 414.5 ms. The radio play group achieved the best result, with a mean motor time of 160.5 ms. The differences between the control group (168.5 ms) and the music group (169 ms) were minimal. The results are presented in Table 3.

Due to significant deviations from a normal distribution, a Kruskal–Wallis Test was carried out as a non-parametric procedure for further statistical analysis, which showed χ² = 3.211, 2 d.f., *p* = 0.201 (average reaction time raw value) and χ² = 0.23, 2 d.f., *p* = 0.988 (mean motoric time raw value). The null hypothesis can therefore not be rejected for all three parameters, as the *p*-values are above 5 percent (*p* = 0.201 > 0.05 for the mean reaction time, *p* = 0.988 > 0.05 for the mean motor time).

### 3.2. Cognitrone Test

In the Cognitrone test, the music group with a mean time of 2.06 s scored better in the “correct rejection” than the radio play group with 2.23 s and the control group with 2.28 s (see Table 4). Only minor differences without significance were found.

Due to significant deviations from a normal distribution, a Kruskal–Wallis Test was carried out as a non-parametric procedure for further statistical analysis, which showed χ² = 5.157, 2 d.f., *p* = 0.076. The main variable examined (mean time “correct rejection”) showed no significant difference between the three groups. The null hypothesis could not be rejected because the *p*-value was above 5 percent (*p* = 0.076 > 0.05 for the mean time “correct rejection”).

### 3.3. Line Tracking Test

The orientation performance was compared in the median of the variable score, which includes the factors accuracy and speed, and was better in the control group (score 17) than in the radio play and music group (both score 16). The results are presented in Table 5.

There was no significant difference between the three groups. The null hypothesis could not be rejected for the parameter score, as the *p*-value was above 5% (*p* = 0.298 > 0.05 for the score raw value). Due to significant deviations from a normal distribution, a Kruskal–Wallis Test was carried out as a non-parametric procedure for further statistical analysis, which showed χ² = 2.425, 2 d.f., *p* = 0.298. The null hypothesis could not be rejected for the parameter score because the *p*-value was above 5 percent (*p* = 0.298 > 0.05 for the score raw value).

### 3.4. Determination Test

In the determination test, the control group with a median of 0.76 s was faster than the radio play and music group (both 0.77 s).

There was no significant difference between the radio play, music and control groups (see Table 6). Due to significant deviations from a normal distribution, a Kruskal–Wallis Test was carried out as a non-parametric procedure for further statistical analysis, which showed χ² = 0.932, 2 d.f., *p* = 0.627 (Mode Action Median Reaction Time Raw Value) and χ² = 3.068, 2 d.f., *p* = 0.216 (Mode Reaction Median Reaction Time Raw Value). The null hypothesis can therefore not be rejected as the *p*-values are above 5 percent (*p* = 0.627 > 0.05 for the median of the reaction time in the action mode and *p* = 0.216 > 0.05 for the median of the reaction time in the reaction mode).

### 3.5. Tachistoscopic Traffic Perception Test (TAVTMB)

Attention performance was better in the music and radio play group, with a median of 80 percent correct answers in both groups compared to the control group (67 percent). The results are presented in Table 7.

Due to significant deviations from a normal distribution, a Kruskal–Wallis Test was carried out as a non-parametric procedure for further statistical analysis, which showed χ² = 1.570, 2 d.f., *p* = 0.456. The null hypothesis could not be rejected for the examined parameter, as the *p*-value was above 5 percent (*p* = 0.456 > 0.05).

### 3.6. Selective Perception of Radio Play Content and Music

The data from the questionnaires were evaluated and presented in bar and line charts. Wrong answers and those marked with “don’t know” were added together, as these were considered to be incorrectly answered. The six questions on the radio play were answered correctly by 31 percent of the test persons and incorrectly or insufficiently by 69 percent. It can be seen that the questions were answered much more incorrectly over time. For example, 21 out of 30 respondents answered the first question correctly, whereas no respondent answered the last question correctly. This may have been due to the increasing complexity of the questions, but also to the increased test complexity or test duration. Figure 4 shows the correctly answered questions over time and Table 8 shows the presentation of correctly and incorrectly answered questions on the radio play.

After completing the test, the music group had to answer questions about the music they heard. The results are displayed in Table 9 and graphically in Figure 5.

The results suggest that the music is well known to most respondents, probably also due to the fact that very popular music by the Beatles was selected. The majority of the test persons had already heard the songs presented or thought they knew some of them. However, less or no attention was paid to the content of the lyrics during the test. When asked how pleasant or unpleasant the test subjects felt about the radio play or music during the test, the radio play group gave a mean value of 6, and the music group a mean value of 2 on a numerical rating scale. This is shown in Figure 6 graphically.

Interesting results were obtained by evaluating the question: “Do you believe that your behavior/reaction would differ significantly if you were to participate in road traffic without any auditory influence (music/radio play thriller)?” The music group rated the influence of music as lower compared to the radio play group’s rating of the influence of the radio play thriller, which is shown graphically in Figure 7. It is possible that the radio play group was more strongly influenced by the crime thriller heard during the test than the music group.

In order to confirm or reject this statement, the statistical evaluation of the results investigated whether there were differences in the score between the music and radio play groups. In both groups, the distribution of the score values was found to be the same. With a *p*-value of 0.061, the null hypothesis was maintained. The results show that the influence of the music was felt to be roughly as strong as the influence of the radio play. In order to find out whether the test persons perceived at all that they were influenced in their reaction behavior, it was then tested whether the score value differed from zero. The null hypothesis had to be rejected here, as the score value was different from zero. The evaluation of the assessment of the subjective perception of the disturbance potential of the radio play or music sound reinforcement showed a significantly increased influence, which was equally pronounced in both groups.

In summary, it can be stated that both groups felt influenced in their reaction behavior by the music or radio play to around the same extent.

## 4. Discussion

In the five tests which we carried out, no significant differences in the results between the control, radio play and music groups could be measured overall. A similar study also found no significant differences in the responses to relaxation music compared to a control group during a driving simulation [5]. Other authors found that fast music [8] and loud music [10] significantly increased risk taking and accident frequency. Numerous studies investigated driving behavior under stress induction, which was triggered in different ways. Music had a negative effect on braking behavior when the difficulty of the track was increased and even had a positive effect on stopping distances on easy tracks [4]. Another work had induced the stress by solving a cognitive test. No worse driving behavior could be observed compared to a group without parallel listening to music [24]. However, in order to prove the effects of stress on reaction behavior, many factors must be taken into account. For example, introverted persons already have their optimal activation level without background stimulation and work worse when additionally influenced by background music. Extroverted people, on the other hand, can achieve a better level of activation through additional stimulation and work better qualitatively and quantitatively [25,26,27]. Other studies have also found a significant improvement in performance under the influence of music [25]. The degree of difficulty of the main activity, the length of the background music and the individual personality played a role here [28]. The difficulty of completing the primary activity, in the studies examined here—namely active participation in road traffic—seems to play a role in reaction and concentration behavior.

In our study, we found a subjectively higher level of stress among participants in the final survey when they were asked to participate in a radio play. They also believed that their reaction behavior was actually impaired by it, to a greater extent than the music group.

Age and experience can also have an influence on reaction behavior. For example, experienced road users are less distracted than novice drivers [29]. The driving routine therefore plays an important role. In comparison, the driving routine was not considered separately in our study. However, it seems to play an important role as a factor for possible differences in performance in the studies. In the studies of different age groups, it was observed that experienced drivers reduced their driving speed accordingly when the difficulty of the driving simulation increased (e.g., a pedestrian suddenly crossing the road). No reduction in driving speed was observed among the younger drivers, which led to an increased risk of accidents. On average, all test persons solved the tasks equally well [30]. In our study, we did not conduct an age-specific examination, but it seems that age can play a role.

A further working group investigated whether conversations or the solving of simple tasks during monotonous driving distances can have a certain activating effect on the brain to simultaneously master the driving task [6]. It was concluded that simple conversations during a monotonous journey have a certain activating effect and can therefore increase concentration. However, this effect only occurs during longer monotonous driving times.

The duration of a load also plays a role. In our study, we observed a decrease in the number of correctly answered questions in the radio play over time, which could also correspond to a certain fatigue after completion of all test units. In addition, the test battery represents a stress situation with an increasing degree of difficulty in the determination test, which makes it more difficult to answer the following simple content tasks correctly.

The extent to which the auditory influence contains interesting or uninteresting messages also plays a role. When listening to interesting audio material, extended braking times could be observed in critical situations. Subjectively, however, it is not perceived as more demanding [31]. In our study, the results were homogeneous and without significant differences, but the radio play was subjectively perceived as more disturbing.

The presentation of speech content during the completion of a traffic simulation can lead to a decrease in brain activation in the areas responsible for spatial processing and thus in the concentration performance for the driving simulation in the parietal lobe with simultaneous presentation of speech content [32]. Simultaneous increases in the activation of areas for speech processing in the brain have also been measured [33]. The achievement of cognitive performance while driving in a traffic simulation can therefore lead to a deterioration in driving performance.

Other authors have described an increase in heart rate, blood pressure and cortisol levels when listening to pop music compared to a control group without music [9]. These reactions belong to the vegetative-humoral reactions and can be negative stress factors if they increase more than this.

Recent studies address the general distraction of road traffic participation, particularly through smartphones and multimedia navigation devices. They looked not only at the “blindness” caused by prolonged exposure to mobile phones but also at the acoustic distraction caused by listening to music [34], where listening to music was the activity that caused the most acoustic distraction. A meta-analysis examined 41 reviewed studies of pedestrians, cyclists and car drivers who were distracted by smartphones. This revealed the difficulty of measuring the distraction factor [15]. However, it is generally known that walking, cycling and driving require visual and acoustic perception in order to be able to react to traffic dynamic developments. The topic is considered to be very complex, as the distraction can occur at different levels. As expected, the majority of visually demanding activities resulted in altered reaction speeds in all groups. Of interest were the cognitive influences of demanding telephone conversations among pedestrians, which led to delayed and therefore unsafe crossing of roads and intersections [35]. Headphones are the most common distraction for mobile phone users (19 percent), followed by text messages (8 percent) and telephone calls (5 percent) [36]. Women wrote text messages and made telephone calls more often; men wore headphones more often. Cyclists, on the other hand, seem to use visual compensation strategies when making phone calls [37]. In our study, we could not make any statements about differences in the reaction behavior of cyclists and pedestrians, but the results of these studies are nevertheless interesting in terms of the different mechanisms of perception and processing. 

The effect of traffic diversion with an increase in accident risk changes with age, risk perception and risk behavior [38,39]

In the current literature, a study investigated the influence of audio books on distraction behavior when driving in a driving simulator [17]. A simpler and a more complex course were driven, each with and without audio play noise. A longer braking time was observed during the radio play presentation on more complex driving courses. On the simple driving course, however, the braking reaction time in the case of danger was shortened. Interesting results were observed for persons who had a high OSPAN (Operation Span) value. This was determined in advance for all participants and indicated an increased multitasking ability of the persons. The persons with a high OSPAN score had a shorter braking reaction time in both driving simulations when listening to the audio book. The authors criticized the difficulties in evaluating the results due to the differentiation between mental overload vs. underload under appropriate driving conditions and the individually available cognitive resources.

Based on the results, it can only be assumed that audio books generate improved attention during monotonous driving situations. This depends, among other things, on the radio play presented. Furthermore, the authors are well advised to further investigate the distraction caused by hands-free phone calls in comparison to various audio books. In our study, a children’s audio book was presented. In our study, we had selected a detective story radio play to deliberately increase the distraction factor compared to a simple radio play.

When we compare the study results presented here with our results, we can usually observe a decrease in concentration in a road traffic simulation, among other things, due to additional auditory stimuli. In the available studies, it is mainly the change in loudness, music tempo, the content of the acoustic irradiation and distraction by other disturbing factors that can trigger stress. Even if the results are not always significant, it can be postulated that sound reinforcement has physical and psychological effects on the ability to react when participating in traffic. Depending on the situation and accompanying influences, these can have a positive effect, with improved reaction or vice versa. Other aspects, such as the individual state of alertness, excitement or relaxation as well as the visual-spatial processing, could not be considered in detail in the complex context of our study.

Finally, what is the reason for the lack of significance of our results?

Since the test system used is a valid test system in itself, numerous other possible influencing factors must be taken into consideration. Additional visual influences, the individual activation level of each person (biorhythm, intro vs. extraversion, emotions), the individual traffic situation, previously undiagnosed hearing damage and the parallel performance of additional non-driving activities may play a role. In addition, both driving routine and experience in dealing with computer tests can provide advantages in the execution of tests. Previous studies have shown that a certain routine of quickly grasping new or more complex situations and reacting appropriately and safely is advantageous [24,30]. We have very little influence on other individual differences. Larger groups of test persons can keep the individual sources of error low but never completely ignore them. In a real road traffic situation, the gaze occasionally falls on the MP3 player, the mobile phone display or, in the car, on the on-board computer in order to change settings such as title or volume, which additionally distracts the persons. Many road users are distracted by existing communication and information systems [40,41]. This problem will become increasingly acute as the further development of driver assistance systems promotes autonomous driving. The driver is weighed in safety by technical accessories but must still be able to intervene quickly enough at all times [42]. The cognitive resources of a human being during participation in road traffic are limited; multitasking has its physiological limits. Prevention through education, innovative ideas and stricter laws can help to promote mutual consideration and increase road safety. The consideration of the auditory influence on reaction behavior should generally become more important in future studies, especially in the investigation of the causes of distraction-related accidents. The subject of research on the acoustic influence in road traffic remains crucial due to the increase in electronic means of transport and thus the emergence of new inexperience in dealing with significantly quieter and at the same time faster moving means of transport with a simultaneous general increase in traffic.

## Figures and Tables

**Figure 1 ijerph-17-09226-f001:**
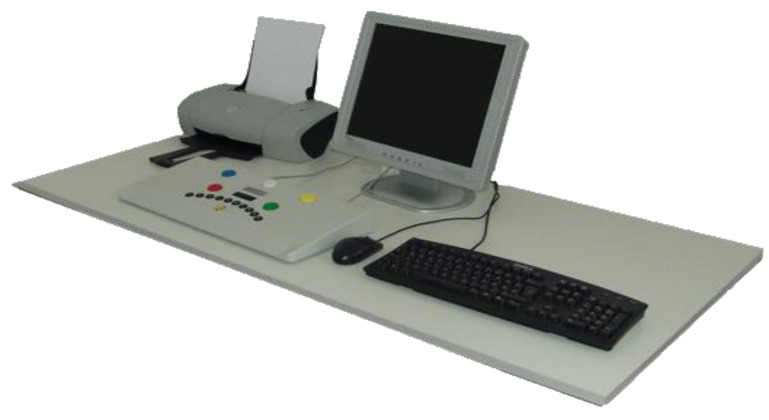
Test person workplace [19].

**Figure 2 ijerph-17-09226-f002:**
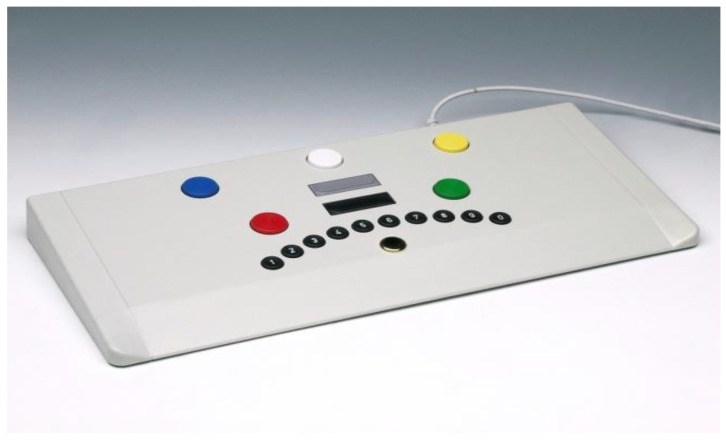
Standard response panel [20].

**Figure 3 ijerph-17-09226-f003:**
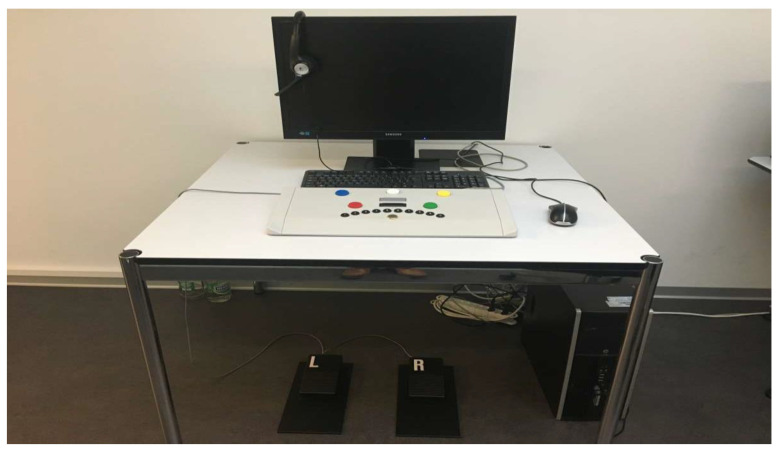
Foot pedals used during the determination test [21].

**Figure 4 ijerph-17-09226-f004:**
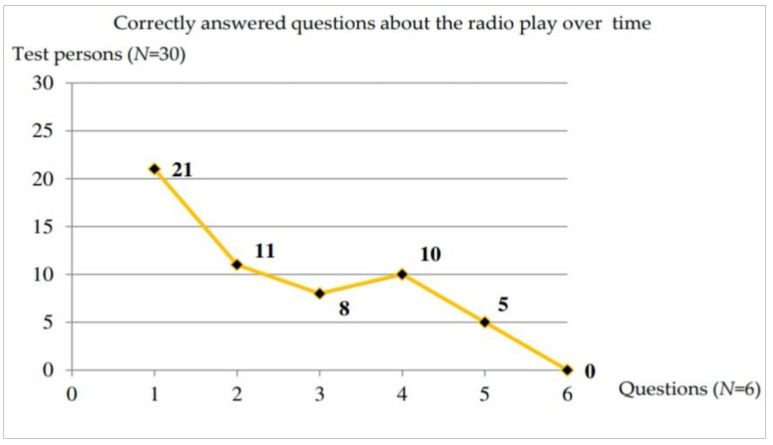
Correctly answered questions to the radio play.

**Figure 5 ijerph-17-09226-f005:**
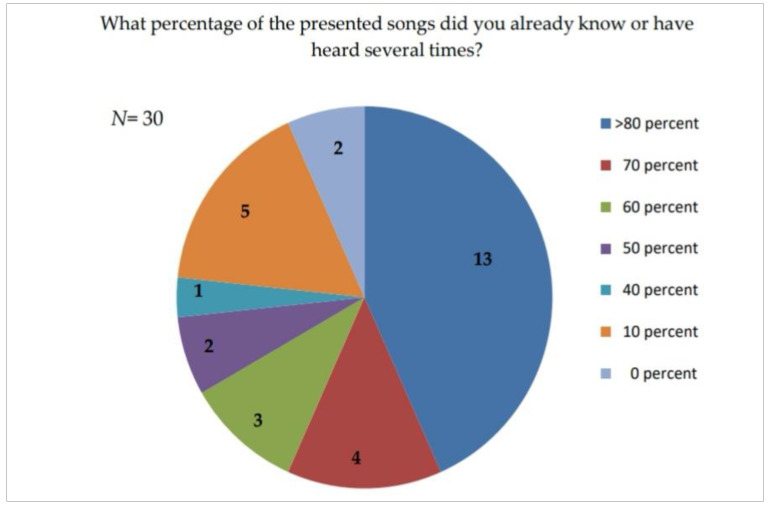
Percentage distribution of answers in the music group.

**Figure 6 ijerph-17-09226-f006:**
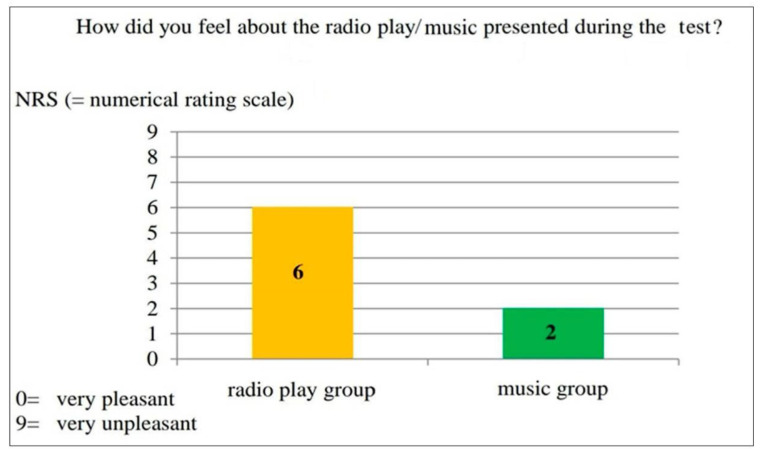
Mean values of the perception of radio play or music during the test.

**Figure 7 ijerph-17-09226-f007:**
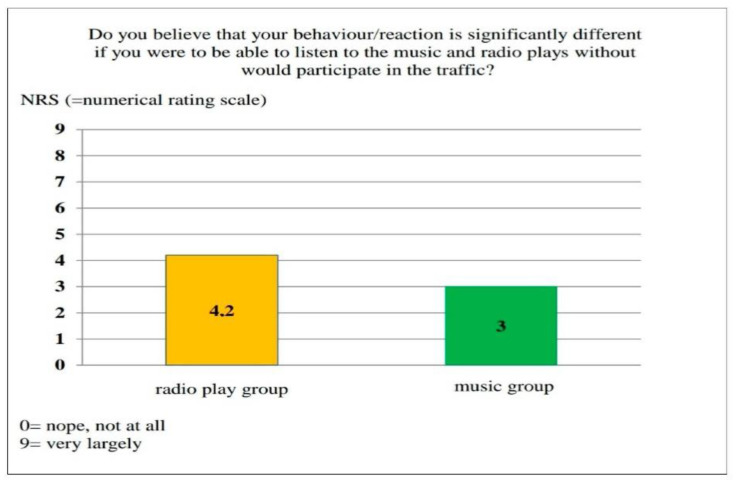
Change in road traffic reaction through music or radio play.

**Table 1 ijerph-17-09226-t001:** Test procedures of the test set FeV Annex 5 No. 2.

Test	Dimensions	Test Forms	Duration (min)
Reaction test (RT)	Responsiveness	S3	6
Cognitronetest (COG)	Concentration capacity	S11	10
Line tracking test (LVT)	Orientation	S3	10
Determination test (DT)	Resilience	S1	4
Tachistoscopic traffic perception test (TAVTMB)	Attention performance	S1	14

**Table 2 ijerph-17-09226-t002:** Implementation and evaluation of the individual tests.

Reaction test	Implementation	Reply field as input medium Animated instruction and practice phase Color stimuli and/or acoustic signals Reaction key is only pressed if relevant stimuli are presented Finger is placed back on the rest button in the connection Headphones for undisturbed stimulation
	Evaluation	Interpretation of the main variable, Average reaction time (msec) Time between a stimulus and the beginning the mechanical movement reaction (the Exit the rest button) Reactions are subjected to a Box–Cox transformation (optimal distribution of reaction times) Distortion of arithmetic averages with extended reaction times and false significances is to be achieved Average motor time (msec) Speed of the movements is determined Should be lower than the reaction time Prolongation may indicate psychomotor disorder; too much shortening (<50 ms) may indicate incorrect execution
Cognitronetest	Implementation	Test includes 60 stimuli Allows statements about attention and concentration 24 hits and 36 correct rejections are the maximum values; Keyboard as input medium Animated induction phase and error-sensitive practice phase test form with free processing time Abstract figure is compared with original in terms of identity After entering the answer, systematically move on to the next task Test form with fixed processing time Reaction only necessary if figure is identical to template After the time has expired, move on to the next task, skip or go back not possible
	Evaluation	Main variable: Mean time “correct rejection” Selective attention in terms of energy to maintain a certain level of accuracy At least 85 percent of the required stimuli (hits) and 85 percent of the unrequested stimuli (correct rejection) must be correctly assessed The 85 percent criterion expresses a personal pace of work It is automatically detected by the program The “Correct rejection” is the indicator for the ability to concentrate and the automatically output standard value can be interpreted as
Line Tracking Test (LVT)—Test Form S3 (screening form)	Implementation	Measuring complex perceptual situations Simple optical structures in a relatively complex environment are pursued in a targeted manner, unaffected by time pressure Measurement of selective attention in the visual field Measurement of concentrated targeted perception Test includes 18 pictures, 18 correct answers are possible Combined instruction and practice phase
	Evaluation	Main variable score measures speed performance and quality of performance High score means fast and accurate perception performance in terms of gaining an overview The factors accuracy and speed are included in the variable score
Determination Test (DT)—Test Form S4 (Hanoverian form)	Implementation	Measurement of complex multiple stimuli—multiple reaction stress, attention and reaction speed The behavior under varying degrees of psycho-physical stress is investigated [22] Stress lies in the continuous, preferably sustained, rapid and varying response to rapidly changing stimuli. Stress occurs when a highly motivated individual is unable to find an appropriate response to a constellation of extreme stimuli [23] Reactive resilience, attention and reaction speed are measured while continuously demanding fast and different reactions to rapidly changing optical and acoustic stimuli Test duration is 300 s The shorter the reaction time, the more (correct) actions are possible in principle Depending on the stimulus-response mode, the variables median reaction time, number of correct (timely, delayed), number of incorrect, number of omitted reactions and number of stimuli are evaluated
	Evaluation	*Median reaction time in subtest 1 mode Action* Main variable of the test form. In action mode, the median reaction time is the given subtest duration divided by the number of correct reactions People with a high value are very well able to react very quickly *Median reaction time Subtest 2-mode reaction* Persons with a high value in this variable and also in the variable “on time” are better than average at reacting adequately (correctly) to simple tasks over a longer period of time under stress
Tachistoscopic Traffic Perception Test (TAVTMB)—Test Form S1	Implementation	Test to check the optical perception performance with regard to gaining an overview and observational ability by means of short presentations (1 sec.) of traffic situations After an instruction phase with two test images, the test person is shown 20 images with a presentation time of 1 s The respondent is then asked to indicate what he or she saw in the picture by answering five different questions The test takes around 10 min.
	Evaluation	Main variable, overview: the characteristic value “gaining an overview” is a measure of the accuracy and speed of visual situation recognition and observational ability. A high percentile rank indicates a well developed ability to grasp situations quickly and accurately. Test value thus expresses most clearly the ability to perceive and the speed of perception

**Table 3 ijerph-17-09226-t003:** Descriptive statistics, reaction test.

Group	*N*	RT (Response Time) Middle Response Time Raw Value, Median (ms)	RT (Response Time) Middle Motoric Time Raw Value, Median (ms)
Listening group (HS)	30	437.5	160.5
Music group (MS)	31	422.0	169.0
Control group (KG)	30	414.5	168.5

**Table 4 ijerph-17-09226-t004:** Descriptive statistics, Cognitrone test.

COG Average Time Correct Rejection (s) Raw Value		
Group	*N*	Median (s)
Listening group (HS)	30	2.23
Music group (MS)	31	2.06
Control group (KG)	30	2.28

**Table 5 ijerph-17-09226-t005:** Descriptive statistics, line tracking test.

LVT Score Raw Value		
Group	*N*	Median (Score)
Listening group (HS)	30	16.0
Music group (MS)	31	16.0
Control group (KG)	30	17.0

**Table 6 ijerph-17-09226-t006:** Descriptive statistics, determination test.

Group	*N*	Modus Action Median Reaction Time Raw Value, Median (s)	Modus Reaction Median Reaction Time Raw Value, Median (s)
Listening group (HS)	30	0.77	0.69
Music group (MS)	31	0.77	0.66
Control group (KG)	30	0.76	0.67

**Table 7 ijerph-17-09226-t007:** Descriptive statistics, Tachistoscopic Traffic Perception Test.

TAVTMB Getting an Overview Percentage		
Group	*N*	Median (Score)
Listening group (HS)	30	80.0
Music group (MS)	31	80.0
Control group (KG)	30	67.0

**Table 8 ijerph-17-09226-t008:** Presentation of correctly and incorrectly answered questions to the radio play.

Question	Correct (*n* = 30)	Correct (Percent)	Wrong (*n* = 30)	Wrong (Percent)
1	21	70	9	30
2	11	37	19	63
3	8	27	22	73
4	10	33	20	67
5	5	17	25	83
6	0	0	30	100

**Table 9 ijerph-17-09226-t009:** Representation of the distribution of responses to the music content.

Did You Pay Attention to the Music Content during the Test?		
	(*n* = 30)	Percent (%)
Yes, very precisely	3	10
rather less	19	63
not a bit	8	27

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
