# Peer review of "The Influence of Different Auditory Stimuli on Attentiveness and Responsiveness in Road Traffic in Simulated Traffic Situations"

_ijerph, 2020, doi:10.3390/ijerph17249226_

Round 1

Reviewer 1 Report

The article focuses on improving road safety by examining the effect of various auditory stimuli on driver attentiveness and responsiveness The article title adequately reflects the content. In the abstract, the author briefly adduces the article's essence. The abstract is sufficient both on the text volume and for understanding the content of the study. The keywords correspond to the article content.

The introduction contains a clearly described goal and tasks of the study, as well as the topic's actuality, which makes it possible to understand and evaluate the ideas, research methods and results, which show the author. The main research directions in this area, as well as unresolved issues are shown.

The article has a logical structure that allows you to form a clear idea of the solution of both theoretical and practical tasks performed by the author during the study. This is reflected through the names of sections and subsections. The author uses scientific and mathematical methods appropriate to the subject under study. The article was prepared in accordance with the instructions for the authors, consistent with the topic that it explores and publishes. According to the study's results, the author gives conclusions, supported by figures and diagrams related to the field of research. During the discussion, the author develops and comments on the meaning of the results obtained and their significance. In conclusion, the author briefly and clearly presents the scientific statements regarding the subject being studied, and gives recommendations for possible further research. Tables and figures are of good quality. References and bibliographies provide insight into the issues at hand. The article has a clear presentation and coherent structure. The possibility of proposed method practical application is shown. In our opinion, the article corresponds to the topic “road safety” and corresponds in type to Preliminary communication.

General conclusion.

The article is devoted to a topical topic, since road safety issues in the context of increasing motorization, improving the vehicles' technical characteristics and complicating traffic conditions can be priorities, especially for young drivers. The authors study the impact of different auditory stimuli on attentiveness and responsiveness in extreme traffic situations. Based on studies in three groups, differing in auditory impact, the authors draw conclusions about which of the impact types has the greatest influence on the movement participant response time. The authors point out the need for preventive work, especially among young people, due to the growth of individual vehicles such as scooters, as well as mobile communication devices. In addition, the authors indicated the direction of further research. The article was prepared in accordance with the instructions for authors, corresponds to the topic that it researches and publishes. In our opinion, the article corresponds to the topic “road safety” and corresponds in type to Preliminary communication. In our opinion, the article can be published without significant modifications.

Author Response

I would like to thank you for the positive comments in your review report.

I hereby send you the edited Version.

Wolfgang Welz

Reviewer 2 Report

The topic addressed in this paper is worthy of research attention and the paper is reasonably well-written. There are, however, a number of corrections and improvements required before I can recommend publication:

Line 40: fatalities fell by 7% compared with what period? If this is comparing one 12 month period with the previous 12 months, then it is not reliably indicative of a trend. What is the trend over 5 or more years?

Line 61: Although they may be instrumental for implementation, I doubt there are many instances where politicians are the main driving force, as initiators, behind road safety reforms and improvements.

Lines 77-82: This passage alludes a potential hypothesis and study aims/objectives. These need to presented much more clearly, possibly under an appropriate sub-heading. 

While there may be some relevance to the overall topic of distraction and auditory stimuli, the first mention of e-bikes and e-scooters comes in the conclusion section and there is no clear link to the study described. This is not the place to introduce new material. It needs to be either developed and introduced earlier in the paper (Intro/Discussion), or omitted.

Line 287: by "weaker", I presume the authors mean "vulnerable" - please edit.

Author Response

line 40:
a decrease of 7% in road deaths compared to 2018 has been recorded. the trend in recent years, after a repeated increase in 2013-2015, is a decreasing number of road deaths from 2016- 2019 onwards.

line61: this was worded somewhat unfortunate. what i wanted to say was that on the political side, decisions are demanded to increase traffic safety due to the increase of traffic and, moreover, of electromobility. there is a lack of laws to increase the safety of all road users due to the future increase of electronically supported transport.

line 77-82:I think you mean the part "We expect descriptive results on the changed reaction and concentration in road traffic
with simultaneous use of audio media via headphones". this can be seen as a working hypothesis, since the publication is part of a doctoral thesis. Better : "we examined the results in reaction behaviour in simulated road traffic situations with simultaneous use of audio media via headphones"

lines 81-82 I have deleted

I will omit the part on electric mobility (e-scooters and e-bikes) on your recommendation.

line 287:I have improved accordingly

Thanks a lot

Reviewer 3 Report

It is a shame that so few of the results were significant, but this research adds to the increasing body of knowledge on the issue and I do believe that we must publish non-significant findings rather than only those that are significant.  However the authors should state in the discussion that they have very little of significance to report and the consider why this should be so when the literature suggests otherwise.

Some things are not reported , or not as well as they might be, for example the U and H statistics and their probabilities, or at least the statistic plus “n.s.” for not significant.  All the significant findings need to have the statistic and probability given. Also the sound levels for the story, music and whether the third condition was in complete silence or other people were talking etc.  Did you conduct any tests by gender- if so what, and what was found?.  Did the participants have preferences for how much noise they would usually drive with, and if so, what sort?  The authors also need to state whether any participants had any hearing problems and that their auditory receptances were all within a normal range. The amount of time spent in the driving simulator { I assume from the description it was such] and what manoevres the participants were required to perform whilst “driving” should also have been documented fully as they may also offer some explanation for the relative lack of significance.

The literature review in this study is weak. There is already quite a  comprehensive literature out there that needs to be consulted and the one here needs to be much more comprehensive in coverage than is presented here. For example what is the effect of auditory information generally  around when driving and what is the effect of the presence of passengers to talk to, etc.  There is also literature on attention switching, which is surely relevant to this study. Also please consider the interaction of auditory and visual stimuli and the distraction issues that this interaction involves, plus the ability of drivers to inhibit visual and auditory distraction.  These are examples, and in one simple search on google scholar I found a lot of relevant articles not considered in this paper. For example, there is a publication on exactly this latter point in Transportation F in 2020 [Karthaus et al] which is one of many that this paper has not addressed.

Also, talking in the conclusion about ensuring electric and other silent vehicles have warning sounds incorporated needs to also consider the impact of pedestrians, cyclists etc  wearing earphones as many do. 

So in summary, there needs to be a better and much more comprehensive literature review, a better stated methodology and sample information, results should be reported properly, and the discussion [and conclusion] needs to reflect, using the existing literature,  as to why these findings here were so largely insignificant.   This is quite a lot of work, but would make the paper publishable.

Author Response

I can agree with your first comment that it is not always the significance of results alone that contributes to growing knowledge.

In the discussion I have incorporated your remarks and asked the question about possible causes of the lack of significance and looked for possible answers.

I can partly answer the statement in the second section. I don't understand exactly what you mean when you say that probabilities of significant outcomes should be given?

The volume for the radio play was individually adjusted so that additional signals were perceived during the test. This was previously adjusted in an instructional phase. The tests were conducted on a gender-neutral basis. The participants were initially asked about their hearing preferences or previously known hearing impairments. However, this was of secondary importance overall. The preferences were difficult to differentiate. Only 2 of the 90 respondents reported pre-existing known hearing impairments. The test was carried out with a computer program to test driving aptitude. I have explained the structure in more detail, the tests are also better explained in the correction.

Based on your critical recommendations, I have increasingly included the literature sources used in the doctoral thesis in the correction and used them for comparison. I am aware that there is a lot of literature. The difficulties actually arise in the consideration of individual criteria. Since participation in road traffic is a complex process and the processing of various stimuli is also very differentiated, it is difficult to expect generalised and significant results. Almost all studies have the problem, when they look at one detail, of fading out the other influences at the same time. Although the concept of distraction has already been included in some accident statistics, a uniform definition of distraction has so far failed to emerge. Distraction can be described as a short-term turning away from traffic and turning towards activities unrelated to traffic. However, this also means a whole range of possibilities for distraction.

I also fail to fully understand your statement about headset-wearing pedestrians and cyclists in comparison with the regulation of warning signals on electric (low-noise) vehicles. Pedestrians and cyclists should always take part in traffic in a concentrated manner, at least up to now the German road traffic regulations do not regulate this.

Road traffic regulations (StVO)

  • 23 Other obligations of drivers

(1) Anyone who drives a vehicle is responsible for ensuring that their vision and hearing are not impaired by the occupation, animals, load, equipment or the condition of the vehicle ...

I hope that I have been able to make a sufficient contribution in the correction to the improved understanding of the study and the results as well as the importance of the work and hope for positive feedback

Reviewer 4 Report

Dear colleagues, 

thank you for the opportunity to review your interesting research. As we strive to improve the safety of road travel for all participants, it is important to understand how different auditory stimuli and distractions can impact risky behavior. The field is not as yet fully developed and the authors aim at addressing this gap with an analysis of the impact of an audio play and pop music. While the experimental design and results are interesting, the framing of the problem, the depth of the analysis, and the presentation and discussion of the results need to be significantly improved before publication.

Content:

  • The distribution of focus of the results seems counterintuitive to me: I understood the main focus to be on risk - with both an objective and a subjective component. However, the objective results are given only 10 lines (161-170), no graphics and hardly any discussion. I would suggest you devote significantly more time and space to these results. The fact that the differences are not statistically significant (to what level?) should be discussed in more depth and contrasted with other studies.
  • Which biases are you addressing in your experimental design? (lines 97-100)
  • I am curious about second-order effects. For example: is there a correlation between the different FEV tests? Is there a correlation between the objective results and the subjective answers in Figure 6? This could be particularly interesting as it may suggest that people are good/bad at estimating their risk behavior and taking corrective action. I think this analysis would allow you to enrich the insights for further research.

Structure:

  • please rewrite the abstract to tighten lines 11-18 and to include a summary of the results of your study
  • Section 4.2 belongs in the introduction / lit review to identify the research gap and formulate your hypotheses. Parts of Section 4.1 belong there as well.
  • Parts of Section 4.1 however should inform Section 2.2. Are you using lines 222-227 to address biases in lines 97-100? If not why not?  
  • A better description and concise but critical review of WTS would be beneficial for the reader
  • I do not know what I am supposed to interpret in Figure 1. What was the question asked? what is the significance of the middle rank? why does the y-axis go negative? 
  • line 94: is the research really 9 years old? if so,you may want to explain in more detail why you are publishing now and why it is relevant.

Style:

  • spellcheck your work thoroughly: (lines 58 (missing space), 72(rd superscript), 140 (Anlage), 173 (interference), 174 (frequency), 230 (i.e.,), 234 (space)
  • edit for clarity conciseness and English syntax throughout, but especially in the abstract and early sections.
  • graph headings need to be clarified (Figures 1 and 2: as far as I can tell applies only to radio plays; Figures 5 and 6: rewrite the title)
  • Figure 6: the numbers do not match the graph 
  • decide between US and UK spelling - you use mostly UK but with some variations (e.g., line 222)
  • line 178: don't understand what that means. Presented to whom? Or are you referring to your current paper? If so, you do not need to specify - show, don't tell!
  • Lines 131-140: FEV is not defined, also capitalized differently
  • line 40: 7% wrt to what number / what year?
  • line 85: you may want to write out "male" and "female"
  • you do not need to write out "line chart", "pie chart", etc. 
  • referencing is confusing - I would suggest you either number in order of citation or alphabetically by author(s) and date and remain consistent in that framework
  • line 109: you may want to translate the name of the play for non-German speakers
  • you may want to de-emphasize the German framing in an international journal (e.g., lines 51-54 vs. lines 65-68 - if I understand correctly the missing data is a German issue but other countries are moving forward; line 54 - "neighbouring countries").  

Author Response

Dear Reviewer 4, thank you very much for your critical comments,

in the correction, i will try to compare the objective and subjective results even better and address the second-order effects. furthermore, i will go into more detail about the tests. in fact, figure 6 aimed to observe possible subjective factors.

i will adapt and change the outline and style according to your recommendations.

to content:

Based on your reference I have the results

is now discussed in more detail and compared with other studies (lines 629-823).

I have eliminated the lines 97-100 that you listed because of misunderstandings.

We suspected second-order effects and therefore tried to make them statistically measurable. I mentioned them later in this paper (lines 606-623). 

to the outline:

I have shortened the summary according to your comment

section 4.2 was integrated into the introduction, and the working hypotheses were specifically listed. parts of 4.1 were also introduced there.

I have eliminated lines 97-100, it was not intended to create a link.

I have explained the Vienna Test System better in the correction

Figure 1 was inserted further back in the corrected version and hopefully makes more sense in the context of the other explanations (lines 606-623). here is now also an explanation why we have compiled these statistics, namely to illustrate it graphically and to find any significant differences, whether and if so, how differently the groups felt influenced by radio play or music.

line 94)The research is actually 9 years old. It is being published at the present time because the topic has a continuing high relevance. The justification is given in the contents of the summary and introduction. 

to style:

I hope that I have corrected any spelling mistakes to my satisfaction.

I have labelled the diagrams to indicate which group was examined.

Figure 6: the graph has been corrected.

I have tried to use UK spelling consistently (line 222 has been corrected).

line 178) I do not quite understand what you mean?

line 131) I have explained FEV in the meaning

line 40) the 7% were put into proportion and also compared in a long-term trend

line 85)male and female were tendered

I have eliminated lines and pie charts

The authors were placed in the order of the quotations as requested.

line 109) the name of the play was translated

the structure was kept a little less national and more international by the inclusion of many additional international studies that i have seen for the work. i think that's what they meant

Round 2

Reviewer 3 Report

The authors have extended the literature review and the discussion, and these are now more or less satisfactory

However the presentation of the results needs improving a lot before the paper can be published.  These are not going to take weeks, it is the presentation here that is the problem. Below are the main issues- I may have missed others but if these principles below are applied whenever they need to be, the paper will look a lot better.

I. had asked for statistics and their p values, and what has been provided is a series of tables, they are very poorly presented for this journal. For example, “table” 5 presents one chi-squared with its df and significance, which properly presented would read. “  In order to test xxx. xxx xxxx a Kruskal-Wallis test was performed, yielding c2=5.157, 2 d.f., p=.076”. [my greek letter "chi" pastes across as c- it should be the letter "c" in "symbol" to get chi].  

Tables 5, 7, and 11 can and should all be presented in one line.  Similarly tables 4, 6, and 10 can be much reduced instead of having a third column with identical values.  For tables 3 and 9, given the non-significance of the findings and the identical degrees of freedom, both these tables could again be each in one sentence, so no table needed.

Figure 4 only has two values, so again could be presented better as just the two percentages in one sentence.

Figure 9 is not presenting the Mann-Whitney as it does not give the significance or the U value, what is there is a cut-and-paste pair of bar charts with middle ranks. 

I would also recommend, as I did before and it has not been done, that the significance levels should be correctly written as n.s..  This is better science than leaving really obviously chance levels of significance in full numerical detail. So whilst not wrong, it looks poor, as if ignorant about the meaning of non-significance.

Author Response

Dear colleague,

Thank you for reading and critically commenting on my publication.

I have tried to deal with all your comments and suggestions.

I have shortened the tables accordingly and tried to make it clearer. I have followed her suggestions for improvement.

Since Figure 9 has created some ambiguity and thus does not contribute to clarity and better understanding, I have eliminated it and also left it at the written explanation.

Finally, I have tried to implement her comments and hope that it is now improved and suitable for submission.

Thank you very much and best regards

Wolfgang Welz

Reviewer 4 Report

Dear colleagues

thank you for the opportunity to review your revised paper. I believe the new version is much improved wrt to the original submission, and it should be ready for publication with a few additional changes.

  1. present the literature review in one section early on in the paper. Now most of Section 4 is, in fact, literature review. Please include that in section 1 and place your research clearly in this context
  2. include your results in the abstract
  3. use "." instead of "," for the decimal point in the tables
  4. Figure 9 "Frequency" is still mispelled
  5. use a table to summarize sections 2.5-2.9
  6. not sure referencing with both (name, date) and [x] helps the reader - you may want to pick one
  7. Furnham et al. (1997) is not in the references
  8. spacing / pagination throughout
  9. use shorter sentences, especially in the abstract

smaller points:

line 26 - underscored comma
line 39 - what is the difference between speed and pace?
line 158 - study - do you mean "stimulus"?
line 173 - font
line 176 - were to undergo = underwent
line 245 - spacing
line 597 - not sure what "is maintained" means
line 604 - interference is mispelled

Author Response

Dear colleague,

Thank you for reading and critically commenting on my publication.

I have tried to deal with all your comments and suggestions.

  1. I have included the literature review in the introduction and have made the results more concrete and shorter in the discussion.

  2. by summary, do you mean the abstract? So far I have been of the opinion that it is not wise to present the results in the abstract, as this would not encourage the reader to continue reading

  3. decimal places were corrected

  4. Figure 9 led to many misunderstandings and ambiguities that I have now explained in the text and eliminated the figure.

  5. a table was created by me.

  6. the referencing was unified/simplified.

  7. Furnham et al (1997) was listed in the bibliography

  8. distance and pagination checked.

  9. sentences were made shorter

To the smaller points:

Line 26 : corrected

Line 39 : corrected

Line 158: no, I didn't mean suggestion - but I have formulated it better now

Line 173: corrected

Line 176: ? with me that would be, I think, line 178... I have reworded it.

Line 245: I cannot find any spacing error there?

Line 597: for me line 601(Thus, the null hypothesis is confirmed for a p value of 0.061

is maintained)... that is a little bit unluckily formulated... better with a p

value of 0.061 the null hypothesis is maintained with a p value of 0.061 the

null hypothesis is maintained

Line 604: unfortunately I cannot understand what you mean?

The term interference potential is, to my knowledge, written as in line 608

Finally, I have tried to implement her comments and hope that it is now improved and suitable for submission

Thank you very much and best regards

Wolfgang Welz